# Effect of Intradialytic Exercise on Hyperphosphatemia and Malnutrition

**DOI:** 10.3390/nu11102464

**Published:** 2019-10-15

**Authors:** Nada Salhab, Mona Alrukhaimi, Jeroen Kooman, Enrico Fiaccadori, Harith Aljubori, Rana Rizk, Mirey Karavetian

**Affiliations:** 1School of Nutrition and Translational Research in Metabolism, Faculty of Health Medicine and Life Sciences, Maastricht University, 6200 MD Maastricht, The Netherlands; nadasalhab@hotmail.com; 2Department of Medicine, Dubai Medical College, P.O. Box 22331, Dubai, UAE; Mona_539@Yahoo.Co.Uk; 3Department of Internal Medicine, Division of Nephrology, University Hospital Maastricht, 6202 AZ Maastricht, The Netherlands; Jeroen.Kooman@Mumc.Nl; 4Medicine and Surgery Department, Parma University Medical School, Via Gramsci 14, 43100 Parma, Italy; Enrico.Fiaccadori@Unipr.It; 5Department of Nephrology, Al Qassimi Hospital, P.O. Box 3500, Sharjah, UAE; Harithmuthana@yahoo.com; 6Institut National de Santé Publique, d’Épidémiologie Clinique et de Toxicologie, (INSPECT-LB) Beirut, Lebanon and Maastricht University, Faculty of Health Medicine and Life Sciences, 6200 MD Maastricht, The Netherlands; 7Department of Health Sciences, College of Natural Health Sciences, Zayed University, P.O. Box 19282, Dubai, UAE

**Keywords:** exercise, hemodialysis, hyperphosphatemia, quality of life, phosphorus

## Abstract

Intradialytic exercise (IDE) is not routinely prescribed in hemodialysis (HD) units despite its potential benefits on patients’ outcomes. This study was the first in the United Arab Emirates to examine the effect of aerobic IDE on hyperphosphatemia, malnutrition, and other health outcomes among HD patients. Participants were chosen from the largest HD unit in Sharjah Emirate for a quasi-experimental intervention with pre and post evaluation. The study lasted for 12 months. Study parameters were collected at baseline, post intervention, and follow-up. The intervention included a moderate-intensity aerobic IDE of 45 min per HD session; intensity was assessed using the Borg Scale. Patients were educated on the importance of exercise. Study outcomes were serum phosphorus (P), malnutrition inflammation score (MIS), quality of life (QOL), and pertinent blood tests. Forty-one eligible consenting HD patients were included in the study. Results at follow-up showed a non-significant reduction in P (*p* = 0.06) in patients who were hyperphosphatemic at baseline, but not in the sample as whole. MIS did not deteriorate throughout the study (*p* = 0.97). IDE resulted in a non-significant increase in the QOL visual analogue scale (*p* = 0.34). To conclude, aerobic IDE for 45 min is safe and could be beneficial, especially for hyperphosphatemic patients.

## 1. Introduction

Hyperphosphatemia, described as the “silent killer” for hemodialysis (HD) patients [1], is prevalent among almost half of this population [2] and is a determinant to decreased quality of life (QOL) [3]. The Kidney Disease Improving Global Outcomes (KDIGO) guidelines (2017) recommend lowering the serum phosphorus (P) levels towards the range that is considered normal for healthy populations. Standard HD regimes of 4-h three times a week cannot remove the entire P load; thus, other serum P controlling interventions are used such as limiting dietary P intake and administration of P-binding medications [4]. However, these techniques provide some risks: the P-restricted diet can lead to protein energy malnutrition [5], and P-binders can cause gastrointestinal side effects [6], which may increase the cost of health care [7]. This is a specific concern because there is no conclusive evidence about P-binder cost-effectiveness as first-line intervention for hyperphosphatemia management [8]. All of which raises the need for adjunct novel effective approaches targeting hyperphosphatemia management, such as intradialytic exercise (IDE). Recently, 2 meta-analyses showed that exercise does not appear to have a significant impact on serum P compared to controls; however, most of the studies included in these reviews were relatively short term [9,10].

IDE has been the focus of many researchers for the last two decades. Orcy et al. (2014) reported an increase in P clearance with IDE among HD patients [11]. Also, other studies showed a decrease in serum P levels with IDE [12,13]. It is hypothesized that exercise increases blood flow and decreases inter-compartmental resistance leading to increased toxin removal through the dialyzer [14]. Moreover, IDE potentially has a positive effect on patient’s QOL [12,15,16,17,18], dialysis adequacy (Kt/V) [15,19], urea reduction ratio (URR) [19,20], C-reactive protein (CRP) levels [21], and functional capacity [22,23]. Finally, in HD patients, body composition is significantly associated with physical functioning and QOL [24]; thus, any attempt targeting muscle and fat distribution should be encouraged.

Despite the existing literature on the positive effects of IDE, it is not yet a routine practice in HD units. In fact, there are neither set guidelines for HD patients to follow, nor the proper counseling on exercise from the medical team [25]. Accordingly, stronger evidence about the safety, effectiveness and cost-effectiveness of IDE are needed to increase the likelihood of implementing it as part of the HD treatment protocol [26,27]. 

The aim of the current study was to assess the effects of a 6-month supervised aerobic IDE program on HD patient’s clinical parameters including serum P, parathyroid hormone (PTH) levels, dialysis efficiency, nutritional status, and QOL. This was the first trial in the United Arab Emirates (UAE) to introduce IDE.

## 2. Materials and Methods

### 2.1. Study Design and Participants

This was a 6-month, quasi-experimental intervention with pre- and post-evaluation design. The study was prospectively registered in the ClinicalTrials.gov (ID: NCT03131804). All procedures performed in studies involving human participants were in accordance with the ethical standards of the Research Ethics Committees of the Ministry of Health and Prevention of the UAE (approval reference number: MOHP/DXB/SUBC/NO-5/2016; approval date: 5th of December, 2016) and of the Research Ethics Committee of Zayed University (ethic application number: ZU15_118_F; approval date: 15th of December, 2015), and with 1964 Helsinki declaration and its later amendments or comparable ethical standards. Informed consent of all eligible patients was obtained before the initiation of the study. The study protocol and baseline results have been published elsewhere [28].

The study recruited patients attending the HD unit at Al-Qassimi hospital, which is the largest HD unit in Sharjah Emirate, UAE, via convenience sampling. The unit nephrologist identified the eligible patients for the study based on the following inclusion criteria: clinically stable, adult, on HD for ≥3 months, free of acute diseases and specific cardiovascular problems (e.g., patients with cardiac pacemaker, or suffering from uncontrolled blood pressure, or symptomatic ischemic heart disease, or arrhythmias, or deep vein thrombosis, or severe dyspnea were not included), not practicing any exercise program at the time of the study, capable of communication, fully aware of the study protocol, able to perform the cycle pedaler, and willing to provide a consent form. The exclusion criteria for the study were patients not meeting the inclusion criteria, dialyzing from a femoral fistula, suffering from severe anemia (hemoglobin: <9 g/dL) [29] and/or uncontrolled diabetes. All eligible patients who gave their written informed consent prior to study participation were included in the intervention. The research team consisted of the primary investigator, along with five research assistants; in addition, the hospital assigned a nephrologist and a physiotherapist for the supervision of the IDE program. 

### 2.2. Study Protocol

The study period was 12 months, and was divided into 3 phases, as follows:

1. Baseline or T_0_ (2 months): This phase comprised the verification of the methods planned for the intervention on 10 randomly chosen patients (results of which were later discarded). This was followed by a one-month individualized patient education on the importance of exercise, specifically IDE. 

2. Intervention or T_1_ (6 months): This phase consisted of an IDE training within the first 2 h of dialysis, using a static pedaler “Pedal Exerciser KD”. The exercise sessions were scheduled for 45 min, 2–3 times per week. Before starting the exercise session, the pedaler was positioned and stabilized on the bed in front of the patients while they lied in their dialysis beds. Exercise modality was as follows: patients started their exercise session with a 2 min warm up, cycling at the lowest resistance of the cycle ergometer [30]; then they continued cycling at their preferred resistance, and were advised to achieve average exertion score of 12 on the Borg Scale of perceived exertion throughout the 6 months intervention [31]. The Borg Scale is used to quantify the intensity of exercise and the effort done by patients; Borg Scale of 12–14 corresponds to a moderate intensity level of exercise [30]. Exercise intensity, duration, and patient’s complaints were recorded for each session. Patient’s blood pressure and pulse rate were monitored and recorded at the beginning of the exercise session and at every 15 min onwards; exercise was not initiated or was stopped if the blood pressure exceeded 200 mmHg systolic blood pressure and/or 110 diastolic blood pressure [32], or was below 100 mmHg systolic blood pressure and/or 50 mmHg diastolic blood pressure. Adherence to the IDE was measured as the number of times patients exercised relative to the number of times they were scheduled to exercise. Patient engaging in ≥60–80% of the exercise sessions were considered to have moderate adherence level, and those engaging >80% were considered to have high adherence level [33]. The study was at all times under the direct supervision of the medical team; in case of any discomfort, the intervention was immediately stopped. The IDE was done in parallel to an ongoing monthly 20-min one-to-one patient education on benefits of exercise for HD patients focusing on dialysis efficiency and hyperphosphatemia management, the importance of IDE, how to initiate a safe workout and integrate it in regular routine. Patients were advised to continue exercising at home after the completion of the program. Written information was handed to the patient at the end of each educational session. The education material was also kept in the HD unit in the form of a standing poster. 

3. Follow-up or T_3_ (4 months): At the end of the intervention, the research team stopped all interaction with the patients, and 3 months post-intervention, assessed them on the study’s outcome measures.

### 2.3. Outcome Measures

The primary outcome was serum P (mg/dL). The P change between the active versus inactive patients was analyzed at T_0_, T_1,_ and T_2_, in addition to a sub-analysis among the hyperphosphatemics (*p* ≥ 5.5 mg/dL) [4] compared with the normophosphatemics. Secondary outcomes included PTH (pmol/L), calcium phosphorus byproduct (Ca × P) (mg^2^/dL^2)^, hemoglobin (HgB) (g/dL), Kt/V, URR (%), malnutrition inflammation score (MIS) adapted from Kalantar-Zadeh et al. (2004) [34], and QOL measured using the EQ-5D-5L questionnaire [35]. The 2:53 PM MIS questionnaire had 10 components, each with four levels of severity from 0 (normal) to 3 (severely abnormal). The sum of all 10 MIS components ranged from 0 (normal) to 30 (severely malnourished). The EQ-5D-5L questionnaire assessed mobility, self-care, usual activities, pain/discomfort, and anxiety/depression. Each question had 5 answer choices. In addition, patients rated their health on a visual analogue scale (VAS) from 0–100 [35]. Adherence and reasons for non-adherence to IDE were assessed and recorded by the research team after each session.

### 2.4. Data Collection

Demographics and blood tests results were collected from the patients’ medical files, and questionnaires were conducted by the research team. Blood tests were collected from patients by the hospital staff before the dialysis session, in a non-fasting state. Data collection time-points were at T_0_ (average of 3 months prior to baseline), T_1_ (end of the intervention phase) and at T_2_ (average of 3 months post intervention).

### 2.5. Sample Size

Sample size was estimated based on a power analysis using serum P as the main outcome variable. It was conducted using the GPower 3.1 software. Assuming a two-sided type I error rate of 5%, a power of 80% and an effect size of d = |0.56| retrieved from Makhlough et al. (2012), the sample size needed was *n* = 34 [13]. We approached all eligible patients (*n* = 57) out of whom 41 consented and were included in the analysis. Considering the participation rate and accounting for the dropout rate, the study design was set to be a quasi-experimental intervention with pre and post evaluation without a control group.

### 2.6. Statistical Analysis

Data were analyzed using the Statistical Package for the Social Sciences (SPSS), version 21 (IBM Corp., Armonk, NY, USA). Results were considered statistically significant at *p* ≤ 0.05. Descriptive statistics were used to summarize baseline characteristics of the sample, whereby categorical data were reported as frequencies and percentages counts, and continuous variables were presented as means and standard deviations. Normality was checked for all continuous variables using the Shapiro-Wilk test. Differences between baseline, post intervention, and follow-up of normally distributed variables were analyzed by the one-way Analysis of Variance (ANOVA) Repeated Measures; those of non-normally distributed data as well as categorical variables, were analyzed by the Friedman Test. We used the pairwise comparison test to compare serum P in active versus inactive patients based on their activity level defined as 20 min of exercise twice a week for 6 months [36].

## 3. Results

Initially, 150 HD patients were assessed for eligibility, 57 were found to be eligible (38%), and 41 consented to participate (28% refusal rate). Over the course of the intervention phase, 6 patients were not satisfied with the cycle used or the IDE itself (dropped at week 10); one patient was transferred to another HD unit at week 12, 2 patients did a transplant at week 6 and 11, and 1 patient passed away at week 10. At follow-up, 1 patient was transferred to another facility. Thus, 30 patients completed the study (Figure 1). 

### 3.1. Characteristics of Patients Who Completed the Study

The patients’ mean age was 48.87 ± 11.29 years, with an equal distribution from both genders, and were mostly Arabs (73.4%) and Muslims (83.3%); 96% of Muslim patients fasted during Ramadan. The primary cause of chronic kidney disease (CKD) was diabetes (53.3%) followed by hypertension (30%). The most prevalent comorbidities were hypertension (93.3%) and diabetes (53.3%). The mean body mass index (BMI) was 25.25 ± 6.71 kg/m^2^. Half of the sample was hyperphosphatemic at baseline. Table 1 illustrates patients’ characteristics at baseline for the ones who completed the study; a full report on the whole sample at baseline has been published elsewhere [28].

### 3.2. Aerobic Exercise

On average, the patients exercised for 140.2 min/month (min 111.2 min–max 166.5 min) on a Borg Scale level of 11.3 (min 10.6–max 12.1) during the 6-months intervention. Monthly patient adherence to the exercise session during the 6 months intervention was as follows: 62% (Month 1), 48% (Month 2), 46% (Month 3), 62% (Month 4/coincides with Ramadan), 64% (Month 5), 63% (Month 6). The mean patient adherence rate to the IDE program was 57.5%. Around 26.6% of the patients were moderately adherent to the program and 20% of the patients were highly adherent. As for the rest, they had varied levels of adherence, which are detailed in Figure 2. The main reasons behind patients not exercising were feeling tired or simply not wanting to exercise, in addition to problems common to HD patients such as leg cramps, leg, and back hurting at night after the exercise, burning feet sensation, shortness of breath, fistula, or catheter problem, patient feeling down, sleeping during the session, or sickness. Only on 5 occasions, the investigators did not initiate exercise because of patient’s high blood pressure. 

### 3.3. Clinical and Laboratory Parameters of Included Patients 

Table 2 shows the laboratory data of patients who completed the study at baseline, after the intervention, and at follow-up. There was no significant difference in all studied parameters in the group as a whole between baseline, post intervention, and follow-up. The most change between baseline and follow-up was seen in the MIS (−18.66%).

The sub- analysis of active versus inactive patients showed that the mean serum P changed from 5.64 ± 1.42 to 5.11 ± 1.21 mg/dL from baseline to follow-up among active patients and from 5.89 ± 2.1 to 5.83 ± 1.79 mg/dL among inactive patient; this difference was not significant in both groups.

### 3.4. Outcome of Patients Who Were Hyperphosphatemic at Baseline

An additional analysis was done to compare the changes in study parameters between hyperphosphatemics versus normophophatemics. In hyperphosphatemic patients, there was a non-significant reduction in serum P mean ± SD from 7.04 ± 1.39 at baseline to 6.08 ± 1.68 at follow-up (*p* = 0.06). The prevalence of hyperphosphatemia was 50% at baseline, 46.6% post intervention, and 46.6% at follow-up. There was no significant difference in the rest of the studied parameters between these 2 subgroups.

### 3.5. Quality of Life (EQ-5D-5L)

There was no statistically significant difference within each of the QOL 5 dimension or in the QOL-VAS (*p* = 0.34) throughout the study; the percentage change between baseline and follow-up in QOL-VAS was 8.93%. The number of patients with “no problem” in most EQ-5D-5L domains (self-care, daily activities, mobility) increased post-intervention (Table 3). 

### 3.6. Adverse Events

During the IDE sessions performed by all patients, only 2.5% of the sessions ended up with complications and these included: 7 incidents of systolic hypertension, 1 incident of diastolic hypertension, 2 incidents of hypotension, and 2 incidents of leg cramps. In these cases, IDE was stopped immediately and when needed, medication was given by the medical team. Furthermore, exercise was mostly discontinued before the completion of the 45 min because most patients reported being tired, or had leg/knee pain. 

## 4. Discussion

To our knowledge, this is the first IDE program to be implemented in the UAE. The 6-month aerobic IDE program showed a tendency towards a positive effect of IDE on serum P levels among the patients. However, no major effects on other parameters such as MIS and QOL could be identified. Around 25% of the patients in the HD unit were eligible and willing to participate in the study. Compared with other single-group interventions of similar nature [12,15,17,19,37,38,39,40,41] our sample number was the largest, except for one retrospective study that gathered 102 patients [20].

In our intervention, the assessment of exercise intensity was subjective; originally, the protocol defined exercise intensity to be at the lowest level of moderate exercise intensity (40% VO_2_R) according to the American College of Sports Medicine guidelines [42]. Unfortunately, most of the patients were on beta-blocker medication, which impedes the increase of the resting heart rate. Consequently, intensity was assessed using the perceived exertion rating (Borg Scale). Thus, our results could have reached significance if intensity was better controlled and if all patients exercised at a moderate intensity at all times. Nevertheless, our results on exercise intensity were close to those reported by a similar study conducted in Canada, where the Borg Scale reported by the patients averaged 10.7 at baseline and 11.8 at 6 months [39]. The average monthly minutes of exercise per patient increased with time during our intervention, while the average patient exertion scale level remained almost constant. This reflects an improvement in the cardiovascular fitness of the participants. As for patient mean adherence level to the IDE, it reached 57.5%, which was bit higher than the results from Bohm et al. (2014) (53%) [39]. Out of the patients who exercised, 26.6 and 20% of the participants achieved moderate and high adherence to the program respectively, which is equivalent to the 26% moderately adherent but lower than the 41% highly adherent patients reported in Germany [33]. 

There was one month in the six-month intervention that coincided with Ramadan—a religious event where Muslims fast during the day and eat after sunset. The experience of another study during this month [43] reported that Ramadan fasters usually are young adults who missed more HD session and had higher serum P level then their older non-fasting counterparts during Ramadan. However, in our sample, during this month, the mean patient minutes of exercise was at its highest, 96% of our Muslim patients chose to fast during Ramadan, adherence to the exercise session was higher than other months, and our patients were considered young (49 years). Thus, IDE may play a role in regulating serum P for the fasting HD patients. 

The current study showed a non-significant reduction in serum P among the hyperphosphatemic patients, in this study, following the KDIGO recommendations, the target serum P level was 3.5–5.5 mg/dL, and hyperphosphatemia cut off point was ≥5.5 mg/dL [4]. This cut off point was also used by Al-Qassimi hospital, where the study took place. Our results can be compared to those of Makhlough et al. (2012), where patient baseline serum P was 7.68 mg/dL and decreased significantly after 2 months of IDE to 5.83 mg/dL [13]. Furthermore, our findings were comparable to other single-group aerobic IDE interventions [15,41], but not to Musavian et al. (2015), who were able to show a significant serum P decrease at the end of this intervention for the general HD population [12]. We assume that hyperphosphatemics are the most that benefit from IDE programs and their number in our sample was small (*n* = 15) which did not allow the results to reach significance. Nevertheless, we should acknowledge that the observed reduction in serum P in hyperphosphatemic patients could be the result of regression to the mean. The lack of a control group impedes us from confirming the direct effect of IDE intervention in this subgroup. It could have been the result of a change in the medication including phosphate binders, or a change in the diet that led to this phosphorus reduction in hyperphosphatemic patients. 

Similar to Musavian et al. (2015), our results showed no significant difference in dialysis efficacy assessed by Kt/V and URR [12]. Other studies that achieved positive significant difference had, either a larger sample size [20], or a preconditioning phase [15] and/or a longer IDE session [19]. In our sample, MIS did not deteriorate, but given the absence of a control group we cannot assess that aerobic IDE helps in preventing the commonly reported deterioration of the nutritional status of HD patients over time [44]. 

In this intervention, the QOL was assessed using the EuroQOl-5D-5L; there was no significant difference within each QOL dimension throughout the study. Our results revealed the clinically positive effect of IDE on most domains of QOL post-intervention; more patients revealed no problem in the self-care, in performing daily activities, and in mobility; fewer patients reported suffering from anxiety, depression, pain, and discomfort. This effect was less evident at follow-up. The reasoning for this may be that the IDE has to be maintained at all times to sustain the improvement of health. Thus, we postulated that the increase in the number of patients reporting slight problems in self-care and mobility at follow-up could be due to the consequences of the normal progression of the disease. In addition, due to the positive rapport created between the patient and the research team, the patients may have felt more comfortable during their HD session, which would have contributed to their overall improvement of their anxiety/depression post intervention. As for the VAS-QOL, it is stipulated that it was over-rated by the study participants who had strong religious beliefs, and their interpretation of sickness was seen as a cause of “fate” and “will of God”; which hindered their ability to rate the domain with objectivity and accuracy. It would have been a good idea to present the VAS without numbers to have more room for improvement. In fact, a more culturally specific QOL instrument should be customized for this population; this was also noted in another study conducted in a Middle Eastern Country [44]. Other research on aerobic IDE has used other QOL instruments in its evaluation. Accordingly, we could not compare the different QOL domains with other studies. However, single groups cycling IDE interventions conducted in various countries, and ranging from 30 to 70 min, have shown significant improvement in QOL total score or specific QOL domains [12,15,17].

Adverse events were encountered during the intervention in 2.5% of the sessions. Our results were comparable to Golebiowski et al. (2012), who had similar setting and recorded complications in 2% of the sessions [17]. Moreover, none of the complications were acute or life threatening, they were the kind that could be managed within the HD unit. 

This study had some limitations; the design was non-randomized, and lacked a control group; thus, clear conclusions could not be drawn. The subjective method used to measure intensity might have influenced the results. In addition, the study was underpowered based on the number of patients who completed the study. Accordingly, future trials with adequate sample size need to be conducted to validate these observations. Last, patient’s diet was no tackled in this study, although renal diet is a cornerstone in the treatment of hemodialysis patients, and could have explained more our results especially in our observations noted in Ramadan. Usually, when people are asked about their diets, they tend to improve or modify it, without realizing. Thus, to prevent any modification of the diet among the patients and its confounding effect on the blood test, diet was not assessed. 

This study had some implications for clinical practice. This intervention suggests that engaging in a 45 min of IDE exercise, 2–3 times per week for 6 months is safe and can be beneficial especially for hyperphosphatemic HD patients. Based on our experience, there are 3 major recommended components for IDE to be successfully implemented in an HD unit: 1) Equipment need to be kept in the treatment room for easy access; 2) Involvement of nursing staff of each HD unit to supervise the exercise program and assign specific coordinators for exercise to help the patient initiate the exercise safely and advise when to stop 3) And most important is the full endorsement of the unit’s head nephrologist for the IDE program. Patient in the HD unit feel fragile and always need the reassurance of their nephrologist to start or proceed with any program. 

## 5. Conclusions

In conclusion, the primary finding of the current study was that a moderate-intensity aerobic intradialytic exercise program appeared to result in a 1 mg/dL decrease in the serum P level among the hyperphosphatemics only; but there was no significant change in the whole group. No significant changes were observed in MIS, URR, and in the VAS-QOL. Future investigations should focus on developing powered controlled trials to validate these observations. Today, we could say that IDE can promise clinical improvement in HD patients. 

## Figures and Tables

**Figure 1 nutrients-11-02464-f001:**
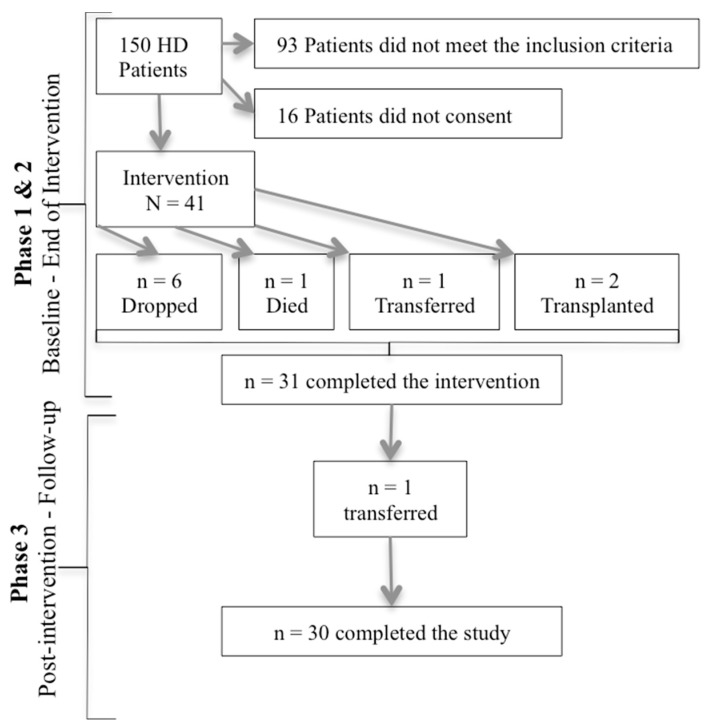
Attrition rate of the study.

**Figure 2 nutrients-11-02464-f002:**
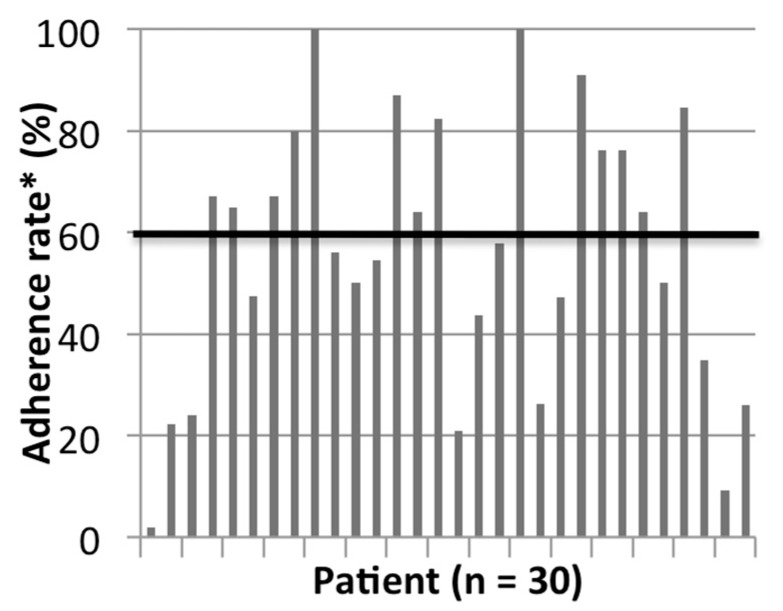
Patients’ adherence to the IDE program. * Moderate adherence: ≥60–80%; High Adherence: >80%.

**Table 1 nutrients-11-02464-t001:** Baseline Characteristics of patients (*n* = 30).

	*n* (%)
Gender, Male	15 (50)
Nationality	
Emirati	5 (16.7)
Non-Emirati—Arab	17 (56.7)
Non-Emirati—Other	8 (26.7)
Primary Cause of CKD	
Diabetes	16 (53.3)
Hypertension	9 (30.0)
Nephritis	3 (10.0)
Others	2 (6.7)
HD Initiation	
<1 year	8 (26.7)
1–4 years	11 (36.7)
>4 years	11 (36.7)
Comorbidities ^†^	
Diabetes	16 (53.3)
Hypertension	28 (93.3)
Cardiovascular disease	4 (13.3)
Others	8 (26.7)
Phosphorus ≥5.5 mg/dL	15 (50) ^‡^
	**Mean ± SD**
Age (years)	48.87 ± 11.29
Weight (Kg)	66.76 ± 17.59
BMI (Kg/m^2^)	25.25 ± 6.71

† Percentages do not sum up due to multiple possible answers. ‡ Valid percentages are reported. Abbreviations: BMI: body mass index; CKD: chronic kidney disease; HD: hemodialysis.

**Table 2 nutrients-11-02464-t002:** Study outcomes of all patients (*n* = 30).

	Mean ± SD Baseline	Mean ± SD Post Intervention	Mean ± SD Follow-Up	*p*-Value/ % Δ
P (mg/dL)	5.78 ± 1.81	5.53 ± 2.14	5.52 ± 1.58	0.56/−4.50
PTH (pmol/L)	49.15 ± 31.65	54.58 ± 38.59	48.80 ± 35.61	0.52/−0.71
Ca × P (mg^2^/dL^2^)	48.96 ± 15.68	46.48 ± 16.48	47.16 ± 14.07	0.54/−3.68
URR (%)	72.93 ± 9.26	73.68 ± 12.38	74.38 ± 10.64	0.47/1.99
Kt/V	1.31 ± 0.1	1.29 ± 0.09	1.28 ± 0.06	0.17/−2.30
HGB (g/dL)	10.74 ± 1.45	11.1 ± 1.66	10.82 ± 1.59	0.52/0.74
MIS	8.2 ± 3.46	7.47 ± 3.39	6.67 ± 1.98	0.16/−18.66
QOL-VAS	63.03 ± 17.65	65.67 ± 19.64	68.66 ± 17.71	0.34/8.93

Abbreviations: % Δ: percentage change from baseline to follow-up; P: phosphorus; PTH: parathyroid hormone; Ca × P: calcium phosphorus byproduct; URR: urea reduction ration; Kt/V: dialysis adequacy; HGB: hemoglobin; MIS: malnutrition inflammation score; QOL-VAS: quality of life-visual analogue scale.

**Table 3 nutrients-11-02464-t003:** EuroQOL 5D-5L Questionnaire (*n* = 30).

		No Problem	Slight- Moderate problem	Severe Problem-Unable	*p*-Value
EQ-dimension		*n* (%)	*n* (%)	*n* (%)	
Self-Care	Baseline	26 (86.6)	2 (6.7)	2 (6.7)	
	Post Intervention	28 (93.3)	2 (6.7)	0 (0.0)	0.368
	Follow-up	13 (43.3)	13 (43.4)	4 (13.3)	
Daily Activities	Baseline	18 (60.0)	7 (23.3)	5 (16.7)	
	Post Intervention	24 (80.0)	6 (20.0)	0 (0.0)	0.070
	Follow-up	24 (80.0)	5 (16.7)	1 (3.3)	
Anxiety/Depression	Baseline	18 (60.0)	7 (23.3)	5 (16.7)	
	Post Intervention	13 (43.3)	7 (23.3)	10 (33.3)	0.111
	Follow-up	20 (66.7)	9 (30.0)	1 (3.3)	
Mobility	Baseline	13 (43.3)	9 (30.0)	8 (26.7)	
	Post Intervention	15 (50.0)	14 (46.7)	1 (3.3)	0.354
	Follow-up	8 (26.7)	17 (56.7)	5 (16.6)	
Pain/Discomfort	Baseline	11 (36.7)	11 (36.7)	8 (26.6)	
	Post Intervention	9 (30.0)	14 (46.7)	7 (23.3)	0.781
	Follow-up	12 (40.0)	13 (43.3)	5 (16.7)	

Abbreviation: n: number of observations.

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
