# Peer review of "Effect of Intradialytic Exercise on Hyperphosphatemia and Malnutrition"

_nutrients, 2019, doi:10.3390/nu11102464_

Round 1
Reviewer 1 Report
Salhab et al. have studied the effect of intradialytic exercise on hyperphosphatemia and malnutrition.
The study question is relevant, the study design is well, and overall it was executed well. The results and discussion presented well.
Author Response
thank you!
Reviewer 2 Report
Quite interesting study about the role of intradialytic phosphorus concentration and patients nutritional status. Few other researchers released papers about it, however results were different compared to this study. Böhm et al. showed in a group of 30 patients that 'acute' exercise may elevate phosphorus concentration and decrease total antioxidant capacity (Böhm J, Monteiro MB, Andrade FP, Veronese FV, Thomé FS. Acute effects of intradialytic aerobic exercise on solute removal, blood gases and oxidative stress in patients with chronic kidney disease. J Bras Nefrol. 2017 Apr-Jun;39(2):172-180. doi: 10.5935/0101-2800.20170022). Adversely, García Testal et al. reported that after 2 hours of exercise during hemodialysis sessions neither lowered postdialysis molecules rebound nor improved dialysis dose (García Testal A, García Maset R, Hervás Marín D, Pérez-Domínguez B, Royo Maicas P, Rico Salvador IS, Meléndez-Oliva E, Molina Aracil J, Murgui Chiva M, Del Pozo Blanco O, Olagüe Díaz P, Fernández Najera JE, Torregrosa De Juan E, Benedito Carrera C, Segura-Ortí E. Influence of Physical Exercise on the Dialytic Adequacy Parameters of Patients on Hemodialysis. Ther Apher Dial. 2019 Apr;23(2):160-166. doi: 10.1111/1744-9987.12762).
However presented study is quite different and shows the effect of exercise (45 minutes per hemodialysis session) during in total 12 months of observation, which gives more information about long-term effect. However:
1) despite tendency to lower phosphorus level after observation period please do not write that a decrease i P was observed (abstract, lines 31-32). Such result is still statistically non significant, however tendency occurs as hyperphosphatemia lowers in hyperphosphatemic group. Similarly: "p-value that approached significance (p=0.06)" (lines 218-219).
2) small language mistakes can be found: "urea reduction ration (URR)" (line 59).
3) Authors wrote that the sample size needed was =34, however at the end only 30 patients took part in this study, do you think the sample size is still appropriate?
4) What do You think about the phenomenon that more patients may reveal slight anxiety/depressive problems (however more showed no problems) after follow up period according to Table 3? Slight problems with mobility and self-care were reported as well.
Author Response
1) Despite tendency to lower phosphorus level after observation period please do not write that a decrease in P was observed (abstract, lines 31-32). Such result is still statistically non significant, however tendency occurs as hyperphosphatemia lowers in hyperphosphatemic group. Similarly: "p-value that approached significance (p=0.06)" (lines 218-219).
Thank you for your comments. We amended the text in the manuscript (line 31, 229-232).
2) Small language mistakes can be found: "urea reduction ration (URR)" (line 59).
We corrected the mistake (line 64).
3) Authors wrote that the sample size needed was =34, however at the end only 30 patients took part in this study, do you think the sample size is still appropriate?
We agree. The study was underpowered based on the number who completed the study. Accordingly, future trials with adequate sample size need to be conducted to validate these observations. We added this explanation to the manuscript (line 336-338).
4) What do You think about the phenomenon that more patients may reveal slight anxiety/depressive problems (however more showed no problems) after follow up period according to Table 3? Slight problems with mobility and self-care were reported as well.
Thank you for highlighting this important matter.This phenomenon could be explained that fewer patients had severe problems in anxiety and depression at follow-up and thus they had either slight problems or no problems at all at the end of the study. This highlights the positive effect of the intervention that was reflected even more at the end of the study. In addition, due to the positive rapport that was created between the patient and the research team, the patients may have felt more relaxed during their HD session, which would have contributed to their overall improvement of their anxiety / depression post intervention. We added this explanation to the discussion (line 320-323). Also, we postulated that the increase in the number of patients reporting slight problems in self-care and mobility at follow up could be due to the consequences of the normal progression of the disease. . We added this text to the discussion (line 318-320)
Reviewer 3 Report
The paper addresses an important issue that has clinical interest. The study design is well thought through and the paper is very well written and presented. However, I feel there are a number of important issues that need addressing:
1) My primary concern is in the interpretation of the results. Reading the abstract and the conclusion, it is easy to conclude that you had a positive result to the trial. In reality, the data shows no statistically significant result and only a p-value of 0.06 in a sub-analysis. It is generally not-advisable to use terms such as "approached significance" "a tendency towards significance" etc. Please can you change these examples in the manuscript (including line 218, 238, 268, 307). You could use terms such as "a non-significant reduction in..."
2) The other major comment is that it would be nice to discuss limitations to the study a little more. This is not necessarily criticism per se, but acknowledging factors that may help explain the results. The striking example to me is diet, which is not really discussed. You talked about ramadan in terms of treatment adherence but not in terms of diet. Also the obvious change in diet in the period immediately after ramadan. What about the potential that the intervention itself has an impact on diet itself, which could counter the hypothesised effect of the intervention?
3) Line 44, replace "HD alone" with "standard HD regimes of four hours three times a week" (it has been demonstrated that extended HD regimes can control phophate)
4) Figure 2 could be remade on a graph with a secondary y axis to allow different scaling for the two measures presented (as it is, changes in the Borg scale are not visible)
5) Would figure 3 be better presented as a histogram?
6) Table 2 - what comparison does the p value refer to?
7) Paragraph 2, page 10, when discussing the sub analysis in the hyper phosphatemic group please consider the phenomenon of regression to the mean.
8) Page 10, para 4. Are there any statistical tests to support the effect of IDE on QOL domains?
9) In limitations of the study, please mention that based on the sample size, your study was underpowered based on the number of participants who completed.
10) Line 317-318. Your data does not back up this conclusion.
Author Response
1) My primary concern is in the interpretation of the results. Reading the abstract and the conclusion, it is easy to conclude that you had a positive result to the trial. In reality, the data shows no statistically significant result and only a p-value of 0.06 in a sub-analysis. It is generally not-advisable to use terms such as "approached significance" "a tendency towards significance" etc. Please can you change these examples in the manuscript (including line 218, 238, 268, 307). You could use terms such as "a non-significant reduction in..."
We thank the reviewer for this comment. We amended the text in the manuscript (line 31, 229-232; 286-287, 358-359).
2) The other major comment is that it would be nice to discuss limitations to the study a little more. This is not necessarily criticism per se, but acknowledging factors that may help explain the results. The striking example to me is diet, which is not really discussed. You talked about ramadan in terms of treatment adherence but not in terms of diet. Also the obvious change in diet in the period immediately after ramadan. What about the potential that the intervention itself has an impact on diet itself, which could counter the hypothesised effect of the intervention?
Thank you for highlighting this important matter. Usually, when people are asked about their diets, they tend to improve /modify it, without realizing. Thus, to prevent any modification of the diet among the patients and its confounding effect on the blood test, diet was not assessed.
We added this clarification to the limitation section in the manuscript (line 341-345)
3) Line 44, replace "HD alone" with "standard HD regimes of four hours three times a week" (it has been demonstrated that extended HD regimes can control phophate)
The change was done on the text as suggested (line 45).
4) Figure 2 could be remade on a graph with a secondary y axis to allow different scaling for the two measures presented (as it is, changes in the Borg scale are not visible)
We agree that figure 2 is not very clear and decided to remove it since it is not adding any value to the manuscript.
5) Would figure 3 be better presented as a histogram?
We presented figure 3 as a histogram. It is now labeled as figure 2
6) Table 2 - what comparison does the p value refer to?
We clarified in line 219 that the p value refers to the significance between the 3 time points: baseline, post intervention and follow up.
7) Paragraph 2, page 10, when discussing the sub analysis in the hyperphosphatemic group please consider the phenomenon of regression to the mean.
In fact, the observed reduction in serum P among the hyperphosphatemic patients could be the result of regression to the mean. The lack of a control group impeds us from confirming the effect IDE intervention in this subgroup. It could have been the result of a change in the medication or the diet that led to this phosphorus reduction in hyperphosphatemic patients. An explanation was added to the text (line 296-300).
8) Page 10, para 4. Are there any statistical tests to support the effect of IDE on QOL domains?
We did not include these results in the article; we added the information to the manuscript (table 3, line 173, 237-238, 312-313).
9) In limitations of the study, please mention that based on the sample size, your study was underpowered based on the number of participants who completed.
We agree. The following text was added to the limitation section: “The study was underpowered based on the number of patients who completed the study. Accordingly, future trials with adequate sample size need to be conducted to validate these observations” (line 339-341).
10) Line 317-318. Your data does not back up this conclusion.
The conclusion has been revised and properly reflects on the article’s presented data (line 358-359).